# High Prevalence and Genetic Diversity of Human Norovirus Among Children Under 5 Years Old with Acute Gastroenteritis at the Dr. Leonardo Guzmán Regional Hospital, Antofagasta, Chile, 2019

**DOI:** 10.3390/v17060794

**Published:** 2025-05-30

**Authors:** Andrea M. Avellaneda, Claudia P. Campillay-Véliz, Daniela C. Reyes, Daniel Herrera, Christian A. Muñoz, Jan Vinjé, Margarita K. Lay

**Affiliations:** 1Department of Biotechnology, Faculty of Marine Sciences and Biological Resources, University of Antofagasta, Antofagasta 1270300, Chile; 2Department of Basic Sciences, Faculty of Sciences, Santo Tomás University, Antofagasta 1243161, Chile; 3Department of Pharmacology, Faculty of Biological Sciences, University of Concepción, Concepción 4070386, Chile; 4School of Medical Technology, Faculty of Health, Santo Tomás University, La Serena 1700000, Chile; 5Clinic Laboratory, Dr. Leonardo Guzmán Regional Hospital, Antofagasta 1240835, Chile; 6Research Center in Immunology and Biomedical Biotechnology of Antofagasta (CIIBBA), University of Antofagasta, Antofagasta 1270300, Chile; 7Department of Medical Technology, Faculty of Health Sciences, University of Antofagasta, Antofagasta 1270300, Chile; 8Millennium Institute on Immunology and Immunotherapy, Department of Biotechnology, Faculty of Marine Sciences and Biological Resources, Antofagasta 1270300, Chile; 9Division of Viral Diseases, National Center for Immunization and Respiratory Diseases, Centers for Disease Control and Prevention, Atlanta, GA 30341-3717, USA

**Keywords:** acute gastroenteritis, human norovirus, genogroups, genotypes, GII.4 variants

## Abstract

Acute gastroenteritis (AGE) has one of the highest rates of morbidity and mortality among children under five years old worldwide. It is estimated that 1.7 billion cases of childhood diarrheal diseases occur annually, leading to up to 443,832 deaths. Approximately 90% of these cases are viral, with human norovirus being the main cause in countries that have implemented rotavirus vaccines. The objective of this study was to describe the prevalence and genetic diversity of norovirus in child outpatients and inpatients under five years old at the Regional Hospital of Antofagasta. From 1 January to 31 October 2019, a total of 121 stool samples were collected to detect the presence of norovirus GI and GII using Taqman™-based real-time RT-PCR. Norovirus RNA was detected in 50 (41.3%) samples, of which 96% were typed as GII.4 Sydney (42% GII.4 Sydney[P16] and 54% GII.4 Sydney[P4 New Orleans]). Furthermore, most (92%) of the positive specimens were from children under two years of age and a majority were detected in April (38%) and May (20%). Our findings highlight the high burden of norovirus in hospitalized children with AGE and the importance of molecular strain surveillance to support vaccine development.

## 1. Introduction

Diarrheal diseases are among the leading causes of morbidity and mortality in children under five years old worldwide, ranking as the third leading cause of death in this age group. According to the World Health Organization (WHO) [1], approximately 1.7 billion cases of childhood diarrheal diseases occur annually, resulting in up to 443,832 deaths in children under five years of age and 50,851 deaths in children aged five to nine years. Viruses are the main causes of infectious acute gastroenteritis (AGE) in children, especially rotavirus, norovirus, and adenovirus [2,3,4,5]. However, with the introduction of the rotavirus vaccine in some countries, there has been a significant decrease in mortality associated with rotavirus [6]. Consequently, norovirus has emerged as the leading cause of viral AGE in the population of all ages worldwide, causing an estimated 70,000–200,000 deaths annually [7]. In the United States, norovirus is estimated to cause 900 deaths, 109,000 hospitalizations, 465,000 emergency room visits, and 2.2 million outpatient visits to health centers [8].

Norovirus disease is generally a self-limiting acute illness characterized by vomiting, non-bloody diarrhea, nausea, abdominal cramps, and mild fever [9]. The incubation period is 12 to 48 h. Some individuals may experience only vomiting or diarrhea. In healthy persons, the duration of symptoms is usually no longer than 48 h [10]. However, young children and the elderly are at an increased risk of developing more severe and prolonged illnesses leading to hospitalization. At least one in six hospitalizations for AGE in children < 5 years in Latin America can be attributed to norovirus [11]. Currently, there is no specific antiviral therapy for treating or commercially available vaccine for preventing norovirus illness [12].

Noroviruses are classified into the family *Caliciviridae* [13]. They are non-enveloped viruses with a positive-sense single-stranded RNA genome, approximately 7.3–7.7 kb in length [10,14]. The genome is organized into three open reading frames (ORFs) with ORF1 encoding a 200 kDa polyprotein, which are post-translational and cleaved in six non-structural proteins: p48, Nucleoside triphosphatase (NTPase), p22, VPg, protease, and RNA-dependent RNA polymerase (RdRp) [10,15]. Noroviruses are currently classified into 10 genogroups (GI-GX) and over 48 genotypes based on the amino acid sequence of the VP1 capsid protein [16]. Recent advances in understanding norovirus evolution, particularly the recognition of recombination events in the ORF1-ORF2 junction region, has led to the development of a dual-genotyping system [17]. This system retains the traditional genotype and genogroup classification while incorporating P-types (over 60 P-types), which are determined by typing a 762-amino acid region of the RNA-dependent RNA polymerase (RdRp) [16].

Global surveillance data indicate that among children under 5 years of age with AGE, the most prevalent genotypes and P-types between 2016 and 2020 were GII.4 Sydney[P16], GII.4 Sydney[P31], GII.2[P16], GII.3[P12], and GII.6[P7] [18]. In Chile, specifically in the Antofagasta Region, no studies have been conducted to identify the norovirus genotypes and variants responsible for AGE in young children. In this study, we describe the prevalence and dual-typing information of norovirus in children < 5 years old diagnosed with AGE at the Dr. Leonardo Guzmán Regional Hospital in Antofagasta during 2019.

## 2. Materials and Methods

### 2.1. Study Participants and Stool Specimen Preparation

Fecal samples were collected from child outpatients and inpatients under five years old diagnosed with viral AGE at the Dr. Leonardo Guzmán Regional Hospital in the city of Antofagasta, Chile, from 1 January to 31 October 2019. The samples were transported to the Advanced Molecular Virology Laboratory of the University of Antofagasta at 4 °C for subsequent storage and processing. Clarified 10–20% (*w*/*v*) fecal samples were prepared in PBS and solids were removed by centrifugation at 10,000 rpm for 10 min and the supernatant was stored at −80 °C.

### 2.2. Viral RNA Extraction

Viral RNA extraction was performed using the QIAamp Viral RNA Kit (Qiagen, Hilden, Germany) according to the manufacturer’s instructions. An amount of 5 µL of the MS2 bacteriophage was added to 135 µL of clarified stool and mixed by vortexing. Then, 560 µL of AVL lysis buffer was added, mixed by vortexing, and incubated at room temperature for 10 min. Subsequently, 560 µL of ethanol (96–100%) was added, mixed by vortexing, and 630 µL of the sample/AVL suspension was added to the QIAamp Mini column (Qiagen, Hilden, Germany), which was centrifuged at 17,709× *g* for 1 min. The columns were washed in 2 wash steps with 500 µL of Buffer AW2 and centrifuged at 17,709× *g* for 3 min. Viral RNA was eluted by adding 60 µL of AVE buffer to the columns and centrifuged at 17,709× *g* for 1 min. RNA extracts were stored at −80 °C until processing.

### 2.3. Detection of Norovirus Genogroups by Real-Time RT-PCR

Viral RNA (3 µL per reaction) was tested for norovirus by multiplex TaqMan real-time RT-PCR assay, which also includes oligonucleotide primers and a probe for MS2 to detect any co-extracted RT-PCR inhibitors [17]. The RT-PCR assay was performed using the Ag-Path One-Step RT-PCR kit (ThermoFisher Scientific, Waltham, MA, USA), according to the manufacturer’s instructions. The oligonucleotide primers and probes have been previously published [17] and include the following: Cog1F (CGY TGG ATG CGI TTY CAT GA), Cog1R (CTT AGA CGC CAT TYA C), and probe Ring1E (FAM-TGG ACA GGR GAY CGC-MGB-NFQ) to detect genogroup I (GI); Cog2F (CAR GAR BCN ATG TTY AGR TGG ATG AG), Cog2R (TCG ACG CCA TCT TCA TTC ACA) and probe Ring2 (Cy5-TGG GAG GGC GAT CGC AAT CT-BHQ2) to detect genogroup II (GII); and MS2F (TGG CAC TAC CCC TCT CCG TAT TCA CG), MS2R (GTA CGG GCG ACC CCA CGA TGA C) and probe MS2P (HEX-CAC ATC GAT AGA TCA AGG TGC CTA CAAGC-BHQ1) to detect MS2. GI and GII primers were used at 400 nM and MS2 primers at 100 nM. GI and GII probes were used at 200 nM and MS2 probe at 100 nM [17]. The total reaction volume was 25 µL. Real-time RT-PCR was performed on the Applied Biosystem 7500 equipment (ThermoFisher Scientific, Waltham, MA, USA) by using the following amplification conditions: reverse transcription for 10 min at 45 °C, denaturation for 10 min at 95 °C, 40 cycles of 95 °C for 15 s, and 60 °C for 1 min.

### 2.4. Amplification of the B-C Region of the Norovirus Genome by RT-PCR

Nucleic acid from norovirus samples positive for norovirus were typed by amplifying the B-C region of the GI and GII genomes using the One-Step RT-PCR kit (Qiagen, Hilden, Germany) as published previously [16]. Specifically, the following primers were used to amplify the B-C region of GI norovirus: MON432 (TGG ACI CGY GGI CCY AAY CA) and G1SKR (CCA ACC CAR CCA TTR TAC A) and MON431 (TGG ACI AGR GGI CCY AAY CA) and G2SKR (CCR CCN GCA TRH CCR TTR TAC AT) for GII norovirus. The RT-PCR conditions used were reverse transcription for 30 min at 42 °C, followed by activation of the Taq polymerase for 15 min at 95 °C, and then 45 PCR cycles of 95 °C, 50 °C and 72 °C for 1 min each, with a final extension of 72 °C for 10 min. Subsequently, the PCR products were visualized by 2% agarose gel electrophoresis, purified, and sent for Sanger sequencing to the Centers for Disease Control and Prevention (CDC), Atlanta, USA.

### 2.5. Sequencing of Regions B-C of the Norovirus Genome

Raw DNA chromatogram files or nucleotide sequences were automatically typed by NoroSurv, using the most recent reference sequences and classification for norovirus [17]. Sequences of representative GII.4 Sydney sequences were submitted to GenBank under accession numbers PV555165 and PV555166.

### 2.6. Statistics

The data were analyzed by groups of patients who were or were not infected (positive and negative for norovirus). Qualitative variables such as age, sex, month of sample collection, treatment and norovirus genotyping were summarized in frequencies and percentages. The association between the groups and each of the categorical variables was evaluated using Pearson’s chi-square test (χ^2^) and Fisher’s exact test depending on each case. Statistical significance was set at a value of *p* < 0.05. SPSS software version 29 was used for all statistical analyses.

## 3. Results

### 3.1. Epidemiological Data

Between 1 January and 31 October 2019, 121 stool samples from children under five years old, who were diagnosed with AGE at the Dr. Leonardo Guzmán Regional Hospital of Antofagasta, were available for testing. Of these, 50 (41.3%) tested positive for norovirus and 71 (58.7%) were negative. Epidemiological data for those patients are shown in Table 1.

### 3.2. Age Distribution of Children with AGE Infected with Norovirus

The average age of 121 patients was 13 months. Of the 50 norovirus-positive samples, 25 were from children under 1 year old, and 21 were from children between 1 and 2 years old (Table 1). However, no statistically significant association was found between the age of the patients and norovirus infection (χ^2^ = 1.968; *p* = 0.58). Of the patients, 33 (66%) were male and 17 (34%) were female (Table 1). Although the percentage of men is almost double that of women, there was no significant association between both variables (χ^2^ = 0.001; *p* = 0.982).

### 3.3. Distribution of Hospitalized and Ambulatory Settings of Children with AGE Infected with NoV

Of all 121 children, 54 (44.6%) were inpatients and 67 (55.4%) received outpatient care (Table 1). Of those with a norovirus infection, 25 (50%) were inpatients and 25 (50%) were outpatients (Table 1 and Table 2).

It is noteworthy that norovirus-positive patients younger than 1 year of age were hospitalized less frequently (36%) than outpatients (64%), and even less frequently than patients between 1 and 2 years of age (52%) (Table 2). This trend was reversed for norovirus-positive children between one and two years of age, who were hospitalized more frequently (52%) compared to outpatients (32%). Despite these percentage differences in distribution, the Fisher exact test (Fischer = 7.1, *p* = 0.06) indicated that these variations were not statistically significant at the 0.05 significance level. Additionally, the low contingency coefficient (0.03) suggests a very weak association between the condition of care (hospitalized vs. ambulatory) and the presence of norovirus in the sample studied.

### 3.4. Seasonal Variation in the Detection Rate of Norovirus

The frequency of stool samples collected from children under five years old with AGE varied throughout the year. Norovirus was detected most frequently in April (19 cases), followed by May (10 cases) and June (7 cases), corresponding to the autumn season in Chile (Figure 1). Notably, the first half of the year (summer and autumn) accounted for the highest percentage of norovirus-positive samples (94%) (Table 1), specifically in April (38%) and May (20%), where the highest percentage of positive cases occurred (Figure 2). On the other hand, the highest number of all AGE cases occurred during the winter months (Figure 2). However, during this period, the presence of pathogens other than norovirus, including rotavirus, *Clostridioides difficile*, and adenovirus, was detected using immunochromatography assays.

### 3.5. Norovirus Genotype Distribution

Among the 50 norovirus-positive samples, 49 (98%) were positive for GII and 1 (2%) for GI. Dual genotyping demonstrated that 54% of the samples were typed as GII.4 Sydney[P4 New Orleans], followed by GII.4 Sydney[P16] (42%), GII.17[P31] (2%), and GI.3[P3] (2%) (Table 3). During the summer months, GII.4 Sydney[P16] was the only type detected, while during the autumn months there was greater variation in circulating genotypes with the presence of GII.4 Sydney[P4 New Orleans] (70.3%), GII.4 Sydney[P16] (27%) and GII.17[P31] (2.7%) (Figure 3).

No significant association was found between infection caused by a specific norovirus type (polymerase: χ^2^ = 3.103, *p* = 0.376 and for capsid χ^2^ = 4.389, *p* = 0.356) and the type of care received (hospitalized/outpatient).

## 4. Discussion

In this study, 41% of the samples from children with AGE tested positive for norovirus, highlighting its importance as an etiologic agent in the pediatric population of Antofagasta. Although our study’s sample size was relatively small due to sample collection limitations, given that the Antofagasta Regional Hospital is an institution focused more on health services than research and does not have sufficient staff for these purposes, our data are in agreement with published global data on the prevalence of norovirus in children under 5 years of age hospitalized with AGE [19,20,21]. Of note, 92% of the norovirus-positive children were younger than two years of age and 50% of all positive cases occurred in children under one year of age. These results corroborate recent publications on the role of norovirus in Chile, including a study of 103 families where one or more healthy children < 24 months of age were followed for 1–2 years to detect AGE, and norovirus was the most frequently detected pathogen [22]. Moreover, in another study in Chilean infants under 2 years of age with AGE, norovirus was detected in 25.5% [21]. Like our study, they also reported more cases in male then in female patients [23]. These results underscore the significant burden of norovirus disease in children under two years of age.

The presence of norovirus in cases of viral AGE in children under 5 years of age showed a strong seasonality toward the Southern Hemisphere’s summer and autumn, with April having the highest percentage of positive cases for this virus (38%). This peak of cases could be linked to the celebration of the Catholic Holy Week in April, in which the high consumption of raw or partially cooked seafood and salads may have led to an increase in infections by gastrointestinal pathogens, including norovirus, in the adult population, who could have transmitted the virus to their children. It is worth noting that 94% of norovirus cases occurred in the first six months of the year, during the summer and autumn seasons. These findings differ from Montenegro’s study [23] in Concepción, a city located in the southern part of Chile, in which norovirus infections in young children were distributed more evenly throughout the year, except in the summer, when rotavirus cases were more frequent. Conversely, the seasonality we detected in our study is similar to that in a study in northeast Brazil [24], in which norovirus was primarily detected during the summer and autumn seasons.

Interestingly, children under 1 year of age whose stool sample tested positive for norovirus were hospitalized less frequently (36%) compared to outpatients (64%), and even less frequently than patients between 1 and 2 years of age (52%). This trend was reversed for children between one and two years of age, who were more frequently hospitalized (52%) compared with outpatients (32%). Although different proportions of positive norovirus cases are observed among age groups according to care condition (hospitalized and ambulatory), the statistical analysis does not provide sufficient evidence to affirm a significant relationship between the need for hospitalization and the detection of this virus in this population. This differs from the findings of a study in Guatemala [25], where 59.2% of patients under one year of age were hospitalized, and 40.8% were outpatients. Likewise, these patients under one year of age were also hospitalized less frequently than those in the same age range that tested negative for norovirus, which is in line with data that one out of six hospitalizations in children < 5 years of age with AGE in Latin America can be attributed to norovirus [11]. The rate of norovirus-positive patients was higher between one and two years of age than under one year of age. This lower number of symptomatic norovirus infections during the first 6 months of life can likely be explained by norovirus-specific IgG and IgA maternal antibodies [26]. In addition, breast milk can inhibit the binding of norovirus VLPs [27], which could also explain the apparent protection from symptomatic norovirus infection in this age group.

The most prevalent genotype in our study was GII.4 Sydney[P4 New Orleans] (54%), which differs from a previous study from Chile [22], where the most frequently detected genotypes were GII.4 Sydney[P16] (23%) and GII.6[P7] (23%). On the other hand, it was observed that only the dual-typed sequences GII.4 Sydney[P16] were detected in summer, while other dual-typed sequences, such as GII.4 Sydney[P4 New Orleans], GII.17[P31], and GI.3[P3], were detected in autumn and winter seasons. However, no significative relationship was obtained between the genotype and the severity of the norovirus infection in children with AGE under five years of age.

To our knowledge, this is the first study reporting a high prevalence of norovirus, as well as genotype information on circulating norovirus strains in children with AGE seeking medical care in the northern part of Chile. Our findings provide a basis for monitoring circulating norovirus genotypes and variants, demonstrating the importance of this pathogen as a cause of AGE in the pediatric population of the Region of Antofagasta, where high incidence rates of this disease have been reported at the national level.

## 5. Conclusions

In conclusion, norovirus was associated with more than 40% of AGE cases in children under 2 years old requiring hospital care at the Regional Hospital of Antofagasta, Antofagasta, Chile, in 2019. Among the genotypes detected, GII.4 Sydney viruses (combined GII.4 Sydney[P16] and GII.4 Sydney[P4 New Orleans]) were the most prevalent. Most of the infections occurred during the first six months of the year (summer and fall). Our data underscore the need for targeted interventions such as vaccines in this vulnerable age group and highlights the need for ongoing molecular surveillance to track changing trends in norovirus genotype distribution.

## Figures and Tables

**Figure 1 viruses-17-00794-f001:**
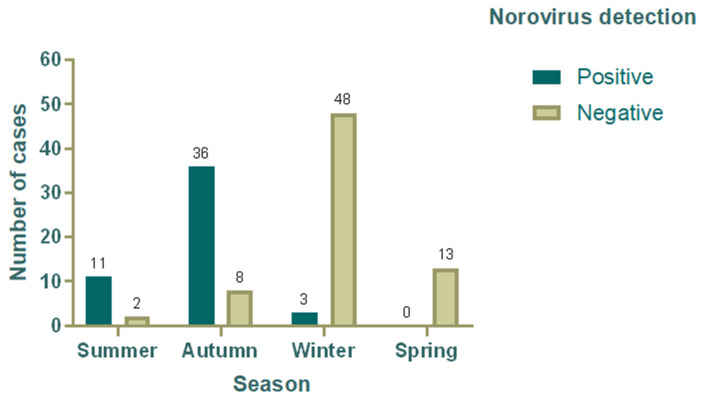
Distribution of positive and negative cases for norovirus by season in children under five years of age.

**Figure 2 viruses-17-00794-f002:**
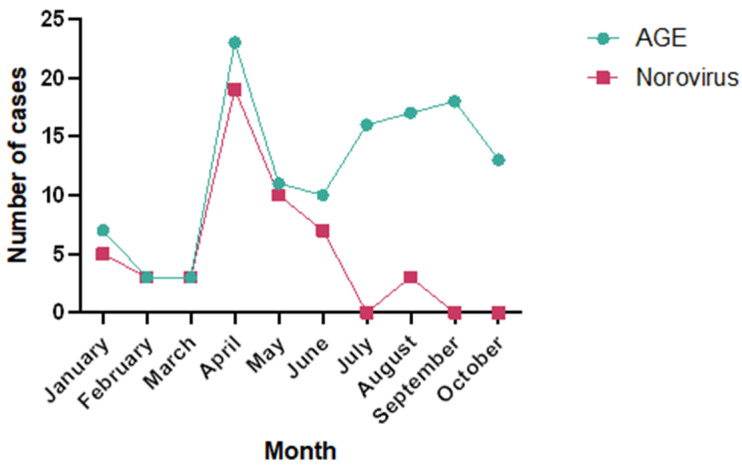
From 1 January to 31 October 2019, at the Regional Hospital of Antofagasta, Chile (*n* = 121). Monthly variation in norovirus cases and AGE cases caused by rotavirus, *Clostridioides difficile* and adenovirus in children under five years of old that attended the Regional Hospital of Antofagasta, Chile, from 1 January to 31 October 2019.

**Figure 3 viruses-17-00794-f003:**
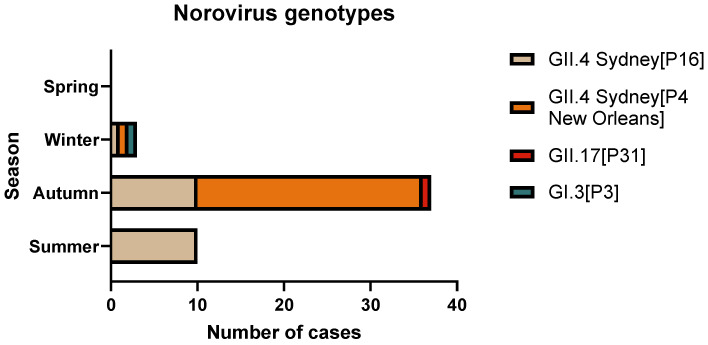
Season distribution of the percentage of norovirus dual-types in children at the Regional Hospital of Antofagasta, Chile, from 1 January to 31 October 2019 (*n* = 50).

**Table 1 viruses-17-00794-t001:** Epidemiological and demographic characteristics of patients under five years old diagnosed with viral AGE at the Dr. Leonardo Guzmán Regional Hospital of Antofagasta during the year 2019.

Characteristic	Total Samples (%)	No. of Norovirus Positive Cases (%)	No. of Norovirus Negative Cases (%)
Gender			
Females	41	17 (34)	24 (33.8)
Males	80	33 (66)	47 (66.2)
Age group (months)			
0 to 12	66	25 (50)	41 (57.7)
13 to 24	47	21 (42)	26 (36.6)
25 to 36	7	3 (6)	4 (5.6)
37 to 48	1	1 (2)	0 (0)
49 to 60	0	0 (0)	0 (0)
Settings			
Hospitalized	54 (44.6)	25 (50)	29 (40.8)
Ambulatory	67 (55.4)	25 (50)	42 (59.2)
Season			
Summer	13	11 (22)	2 (2.8)
Autumn	44	36 (72)	8 (11.3)
Winter	51	3 (6)	48 (67.6)
Spring	13	0 (0)	13 (18.3)
Total	121	50 (41.3)	71 (58.7)

**Table 2 viruses-17-00794-t002:** Distribution of norovirus-positive cases in patients under 5 years old by age group and care condition (hospitalized vs. ambulatory) at the Dr. Leonardo Guzmán Regional Hospital of Antofagasta during the year 2019.

Age Group (Months)	Hospitalized		Ambulatory	
	N°.	%	N°.	%
0 to 12	9	36.0	16	64.0
13 to 24	13	52.0	8	32.0
25 to 36	3	12.0	0	0.0
37 to 48	0	0.0	1	4.0
Total	25	100	25	100

**Table 3 viruses-17-00794-t003:** Norovirus polymerase and capsid genotypes detected in children under five years old at the Regional Hospital of Antofagasta, Chile, 2019.

Genogroups	Polymerase (RdRp) Genotype	Capsid (VP1) Genotype	Frequency (N)	Percentage (%)
GI	GI.P3	GI.3	1	2
GII	GII.P16	GII.4 Sydney	21	42
	GII.P4 New Orleans	GII.4 Sydney	27	54
	GII.P31	GII.17	1	2

## Data Availability

All data are shared in the manuscript.

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
