# Peer review of "High Prevalence and Genetic Diversity of Human Norovirus Among Children Under 5 Years Old with Acute Gastroenteritis at the Dr. Leonardo Guzmán Regional Hospital, Antofagasta, Chile, 2019"

_viruses, 2025, doi:10.3390/v17060794_

Round 1
Reviewer 1 Report
Comments and Suggestions for Authors
This is a classical molecular epidemiology study of norovirus infection as a cause of AGE in the pediatric population in a regional hospital in Antofagasta, Chile. Although the time period is limited (only 10 months) and sample size is small, their findings are significant and agree with published global data on the prevalence of norovirus in children under 5 years of age hospitalized with AGE. The findings provide a basis for monitoring circulating norovirus genotypes and variants, demonstrate the importance of this pathogen as a cause of AGE in the pediatric population in Chile, and the need for a norovirus vaccine for the pediatric population. The manuscript is well written. The data are clearly presented mostly. The discussion and conclusions are concise and sufficient.
There several minor corrections are needed.
Specific comments:
Line 30. Delete “tested for”
Line 47. Move “human” to front of rotavirus or delete it in the sentence.
Line 53. Delete period after illness.
Line 162. The numbers “Of all 120 children, 54 (44.6%) were inpatient and 66 (55.4%) received outpatient care” are not consistent with the numbers in Table 1. It should be “Of all 121 children…
Line 164-166. The data described here are not found in the Table 1. Please add the data under Settings (separate the two numbers of ambulatory vs hospitalized by age groups).
Line 179, 186. The name Clostridium difficile needs to be updated to Clostridioides difficile.
Line 209. Delete period after norovirus before highlighting.
Line 235-236 “Children under one year of age whose stool sample tested positive for norovirus were hospitalized less frequently (36%) compared to outpatients (64%)” This is a very important statement, but the data cannot be found in the Result section. Please add the data to Table 1 under Settings (separate the two numbers of ambulatory vs hospitalized by age groups).
Author Response
Line 30. Delete “tested for”
Response 1: Thank you for pointing this out. We agree with this comment, therefore, we have deleted “tested for”. Page 1, Abstract, line 31.
Line 47. Move “human” to front of rotavirus or delete it in the sentence.
Response 2: Thank you for pointing this out. We agree with this comment; therefore, we have deleted “human” in the sentence. Page 2, Introduction, line 46.
Line 53. Delete period after illness.
Response 3: Thank you for pointing this out. We agree with this comment. Therefore, we have deleted period after “illness”. Page 2, Introduction, line 53.
Line 162. The numbers “Of all 120 children, 54 (44.6%) were inpatient and 66 (55.4%) received outpatient care” are not consistent with the numbers in Table 1. It should be “Of all 121 children…
Response 4: Thank you for pointing this out. We agree with this comment. Therefore, we have corrected the numbers, according to Table 1. Page 4, Results, line 166.
Line 164-166. The data described here are not found in the Table 1. Please add the data under Settings (separate the two numbers of ambulatory vs hospitalized by age groups).
Response 5: Thank you for pointing this out. We agree with this comment. We have, accordingly, added a new table, which now is Table 2. In this new table, it was separated the two numbers of ambulatory vs hospitalized by age groups and performed the respective statical analysis, which is described in Results, in the 3.3. section. Page 5, Results, lines 173-184 and lines 187-195.
Line 179, 186. The name Clostridium difficile needs to be updated to Clostridioides difficile.
Response 6: Thank you for pointing this out. We agree with this comment,
therefore, we have modified the name “Clostridium difficile” to “Clostridioides difficile” in the sentence. Page 5, Results, line 206.
Line 209. Delete period after norovirus before highlighting.
Response 7: Thank you for pointing this out. We agree with this comment. Therefore, we have deleted period after “norovirus” before “highlighting”. Page 7, Discussion, line 260.
Line 235-236 “Children under one year of age whose stool sample tested positive for norovirus were hospitalized less frequently (36%) compared to outpatients (64%)” This is a very important statement, but the data cannot be found in the Result section. Please add the data to Table 1 under Settings (separate the two numbers of ambulatory vs hospitalized by age groups).
Response 8: Thank you for pointing this out. We agree with this comment. We have, accordingly, added a new table, which now is Table 2. In this new table, it was separated the two numbers of ambulatory vs hospitalized by age groups and performed the respective statical analysis, which is described in Results, in the 3.3. section. Additionally, we added new text related to these results to the Discussion section. Page 5, Results, lines 173-184 and lines 187-195; and Page 8, Discussion, lines 289-296.
We would like to thank the Reviewer for his/her time and effort to review this work. We feel that these suggestions have significantly improved our manuscript and hope that the current revised version is acceptable for publication in Viruses.
Reviewer 2 Report
Comments and Suggestions for Authors
Dear Andrea
I have only minor comments.
-
Consistency in Data Presentation: Ensure consistency in the number of deaths reported. The abstract mentions 525,000 deaths, while the introduction cites 494,683 deaths. Please use a consistent figure throughout the manuscript.
-
Reference Updates: Reference 2 is from 2016. It would be beneficial to include more recent publications to support the study's context and findings.
-
Sample Size Justification: The manuscript mentions that 120 stool samples were analyzed. This sample size is relatively small for determining the circulating norovirus types in Antofagasta. Please provide a justification for the sample size or discuss its limitations.
-
Norovirus Type Classification: The names of the norovirus types should be according to the new classification. For example, GI.3 should be written as GI.3[GI.P3]. Please correct the names in the figures accordingly. The text contains GII.4[P31], which should be corrected to GII.4[GII.P31].
-
Sequence Accession Numbers: Sequence accession numbers and the database should be mentioned in the methods section.
Author Response
have only minor comments.
- Consistency in Data Presentation: Ensure consistency in the number of deaths reported. The abstract mentions 525,000 deaths, while the introduction cites 494,683 deaths. Please use a consistent figure throughout the manuscript.
- Response 1: Thank you for pointing this out. We have modified the number of deaths in the abstract to “443,832”, which is the same indicated in the Introduction, now in line 43, accordingly to the reference N° 1. Page 1, Abstract, line 26. We have also double-checked to ensure that the numbers are consistent throughout the manuscript.
- Reference Updates: Reference 2 is from 2016. It would be beneficial to include more recent publications to support the study's context and findings.
- Response 2: Thank you for pointing this out. We agree with this comment. Therefore, we have added 3 recent publications as references to support our study's context and findings. Page 9, References, lines 360-366.
- Sample Size Justification: The manuscript mentions that 120 stool samples were analyzed. This sample size is relatively small for determining the circulating norovirus types in Antofagasta. Please provide a justification for the sample size or discuss its limitations.
- Response 3: Thank you for pointing this out. We agree with this comment. Therefore, we have added a text with the justification for the sample size in the Discussion section. Page 7, Discussion, lines 261-263.
- Norovirus Type Classification: The names of the norovirus types should be according to the new classification. For example, GI.3 should be written as GI.3[GI.P3]. Please correct the names in the figures accordingly. The text contains GII.4[P31], which should be corrected to GII.4[GII.P31].
- Response 4: Thank you for pointing this out. We agree with this comment. Therefore, we have modified the names of the norovirus types according to the new classification (Chhabra et al. JGV, 2019) in the text and in the figures. Results, lines 252-254. Discussion, Page 8, line 307, lines 309-312.
- Sequence Accession Numbers: Sequence accession numbers and the database should be mentioned in the methods section.
- Response 5: Thank you for your comment. Eventhough, it is unusual to submit short sequences to GenBank, to meet your request, we have added the GenBank accession numbers of 2 representative GII.4 sequences (GII.4 Sydney[P4] and GII.4 Sydney[P16], which represent 96% of the sequences detected in is study. Page 3, Materials and Methods, lines 135-136.
We would like to thank the Reviewer for his/her time and effort to review this work. We feel that these suggestions have significantly improved our manuscript and hope that the current revised version is acceptable for publication in Viruses.